# A Dynamic Spatio-Temporal Analysis of Urban Expansion and Pollutant Emissions in Fujian Province

**DOI:** 10.3390/ijerph17020629

**Published:** 2020-01-18

**Authors:** Shen Zhao, Guanpeng Dong, Yong Xu

**Affiliations:** 1Institute of Geographic Sciences and Natural Resources Research, Chinese Academy of Sciences, Beijing 100101, China; zhaos.14s@igsnrr.ac.cn; 2College of Resources and Environment, University of Chinese Academy of Sciences, Beijing 100049, China; 3Key Research Institute of Yellow River Civilization and Sustainable Development & Collaborative Innovation Center for Yellow River Civilization, Henan University, Minglun Street 86, Kaifeng 475001, China

**Keywords:** urban expansion, pollutant emissions, spatial analysis, Fujian province

## Abstract

Urbanization processes at both global and regional scales are taking place at an unprecedent pace, leading to more than half of the global population living in urbanized areas. This process could exert grand challenges on the human living environment. With the proliferation of remote sensing and satellite data being used in social and environmental studies, fine spatial- and temporal-resolution measures of urban expansion and environmental quality are increasingly available. This, in turn, offers great opportunities to uncover the potential environmental impacts of fast urban expansion. This paper investigated the relationship between urban expansion and pollutant emissions in the Fujian province of China by building a Bayesian spatio-temporal autoregressive model. It drew upon recently compiled pollutant emission data with fine spatio-temporal resolution, long temporal coverage, and multiple sources of remote sensing data. Our results suggest that there was a significant relationship between urban expansion and pollution emission intensity—urban expansion significantly elevated the PM_2.5_ and NO_x_ emissions intensity in Fujian province during 1995–2015. This finding was robust to different measures of urban expansion and retained after controlling for potential confounding effects. The temporal evolution of pollutant emissions, net of covariate effects, presented a fluctuation pattern rather than a consistent trend of increasing or decreasing. Spatial variability of the pollutant emissions intensity among counties was, however, decreasing steadily with time.

## 1. Introduction

In 2008, more than half of the world’s population lived in urbanized areas, overtaking the rural population for the first time, and the urbanization rate was projected to reach 66% or so by 2050 [1,2]. The unprecedently rapid urbanization processes taking place in developing countries undoubtedly contribute much to the increased global urban population [3]. Particularly, after the Economic Reform and Opening-up policy, China’s urbanization rate jumped from 17.92% in 1978 to 59.58% in 2018. Decades of fast and stable economic growth were an important driving force of the large-scale rural-to-urban migration process [4].

Explosive population growth combined with limited living space in urban areas caused a series of issues to be solved urgently, among which tackling air pollution and the associated health problems was a priority [4]. Deterioration of air quality in urban areas was mainly caused by emissions of pollutants such as PM_2.5_ (particulate matters with an aerodynamic diameter ≤ 2.5 μm), NO_x_ and SO_2_ [5,6]. A large number of studies had proven the negative effects of exposures to PM_2.5_, NO_x_ and SO_2_ on human health, such as premature mortality [7], lung cancer [8], and cerebrovascular diseases [9]. Pollutant emissions in urban areas were largely driven by anthropogenic processes such as biomass burning, industrial, and mobile sources [10,11].

A growing number of empirical studies have been conducted to examine the relationships between urbanization, economic development, and environmental pollution. For instance, [12] analyzed the influences of urbanization, economic growth, trade openness, financial development, and renewable energy on pollutant emissions in Europe. [13] explored the spatial spillover effects of industrialization and urbanization on pollutant emissions in China’s Huang-Huai-Hai region. Spatial differences in the impacts of urbanization and economic growth on air pollutants in China were further explored by looking at provincial panel data [14,15]. Nonetheless, most of these studies relied on analysis units with coarse spatial scales (either being provinces or prefecture cities), thus ignoring the potential within-unit heterogeneity effects. In addition, the temporal coverage of environmental pollution indicators was most often short (e.g., five to ten years), which might compromise the reliability and accuracy of estimated relationships between urban expansion and environmental pollution. Furthermore, few studies explicitly took into account spatial correlation and temporal dynamics simultaneously when modeling urbanization and pollution data under study. However, it is a well-known fact that ignoring spatio-temporal correlations could lead to unreliable statistical inferences on relationships between covariates under key research interest [16,17,18]. The distributions of urbanization and environmental pollution were rarely uniform or even over space. Instead, they often exhibited evident spatial characteristics such as clustering patterns, partly because of the spatial sorting and clustering of economic activities and populations [19]. Addressing the aforementioned issue needs an interdisciplinary perspective [20], high-quality and integrated data sets linked from various sources such as remote sensing and satellite images [21], and appropriate methodologies capable of dealing with complex underlying spatial and temporal effects of the data.

This paper attempts to offer a reliable estimate of the relationship between urban expansion and pollutant emissions by carefully addressing the above issues. It first uses pollutant emissions data with fine spatial resolution and long temporal coverage (~20 years), which was compiled by using a bottom-up approach and had been rigorously tested in previous studies [22,23]. In addition, we compiled two urban expansion indicators exploiting various remote sensing and satellite data sources such as the nighttime light data (NTL). New instruments of urban expansion using NTL and satellite images, moving beyond the traditional measures of urbanization from administrative statistical data, have been proposed and employed in environmental studies. A comprehensive review on this strand of literature was provided by [21], and the economic and statistical rationales of these indicators were offered in [24]. Finally, this study built a Bayesian spatio-temporal autoregressive model to estimate the relationship between urban expansion and pollutant emission, explicitly and flexibly modeling spatial correlations and temporal dynamics underlying the data under investigation. Although we took a case study of Fujian province of China, the research methodology and design could be readily applied to other data sets. With respect to the key empirical results, we found a significant relationship between urban expansion and pollution emission: urban expansion in Fujian province during the two decades from 1995 to 2015 significantly elevated PM_2.5_ and NO_x_ emissions intensity. This result was insensitive to two different measures of urban expansion and held after controlling for potential confounding effects.

The remainder of this paper is structured as follows. Section 2 describes the study area, data, and research methods. Section 3 and Section 4 present findings and discussions from our descriptive analyses and statistical modeling. Section 5 concludes with a brief summary of the findings.

## 2. Materials and Methods

### 2.1. Study Area

Our study area is the Fujian province, located on the southeast coast of China (28°30′–28°22′ N, 115°50′–120°40′ E). It is adjacent to Zhejiang province in the northeast and borders Jiangxi province in the west and northwest and Guangdong province in the southwest (Figure 1). The land area of Fujian province is 1.24 × 10^5^ km^2^ with a population of about 37 million in 2010 [25]. There are 85 counties constituting Fujian province, serving as our analysis units. The average land area of the counties is about 1436 km^2^ with a standard deviation of 1028 km^2^ whilst the average population is 434,000 with a standard deviation of 319,000 [25].

### 2.2. Pollutant Emissions Data

Environmental pollution impacts of urbanization were measured by using pollutant emissions instead of pollution concentrations in this study. First, pollutant emission data were compiled with fine spatial resolution (~0.1° by 0.1°) and relatively long temporal coverage. Emissions of various pollutants such as PM_2.5_, NO_x_, SO_2_, and TSP were compiled by using a rigorous bottom-up approach based on energy consumption and source-specific emission factors. Details on source information and the calculation and calibration procedures were provided in [22,23]. Pollutant emissions data were made publicly available by the School of Environmental Sciences at Peking University (http://inventory.pku.edu.cn). Second, high space–time resolution emissions data can serve as an important input into the GEOS-Chem atmospheric transport model to derive high-resolution and spatio-temporally consistent pollutant concentrations [26,27]. Finally, official air pollution monitoring data did not have the necessary temporal coverage required by the present study. Accuracy of monitoring data could also be compromised by various spatial interpolation operations when deriving areal pollution summaries from monitoring station readings. Monthly gridded pollutant emissions data in 1995, 2000, 2005, 2010, and 2015 covering the study area were downloaded and aggregated to annual cumulative measures of pollutant emissions for counties. As county boundaries and grids were not compatible geographies [28], a standard GIS areal weighting approach was used to transfer gridded emissions data onto county-scale measurement. We finally obtained the annual pollutant emission intensity of each county by dividing the annual cumulative pollutant emissions by the area of each county.

### 2.3. Remote Sensing Data

Remote sensing-based land use monitoring data were extracted from the Landsat TM/ETM images through the manual visual interpretation method [29]. Land use data in the same years (1995, 2000, 2005, 2010, and 2015) as the pollutant emissions data were downloaded from China’s Resource and Environment Data Cloud Platform (http://www.resdc.cn/), with a spatial resolution of 1 km by 1 km. In this data, land use types were divided into six major categories: cultivated land, forest land, grassland, water area, urban and rural construction land, and unused land. Each major category was further divided into several sub-categories. With this data, we measured urban expansion by forming an urban land development intensity (ULDI) indicator, calculated as the ratio of urban construction land area of a county to its total area during 1995 and 2015. 

As an assurance on the estimated relationship between urban expansion and pollutant emissions, we derived another indicator of urbanization by using the nighttime light (NTL) data available from NOAA/NGDC (https://www.ngdc.noaa.gov/ngdc.html). The NTL data were compiled from DMSP/OLS and NPP-VIIRS with a spatial resolution of 1 km by 1 km in the matched time periods. NTL data have been increasingly used to calibrate indicators of both urbanization and economic development [24,30,31,32,33] (DMSP/OLS and NPP-VIIRS data themselves are not without problems when approximating levels and variations of regional urbanization and economic development: the relatively poor county ad over-saturation of DMSP/OLS data and existences of negative values, extremely high values, and unstable light sources for NPP-VIIRS data [34]). We therefore extracted the NTL data for the study area from 1995 to 2015. Urbanization was measured by dividing the total luminosity of a county by its total area (LD). We acknowledge that remote-sensing-based measures of urbanization focus on the urban land development or expansion. It is less capable of reflecting urbanization processes in other domains, such as population, culture, and lifestyles. It was useful to note that the LD measures of urban expansion were consistent with the ULDI indicator (the Pearson coefficient about 0.8). The gridded ULDI and LD data were used to extract county-level urban expansion measures with the standard GIS areal weighting approach [28].

### 2.4. Geographical Factors

Geographical and locational factors were also incorporated in our pollutant emission models to capture potential confounding effects. Geographical variables included the greenness of each county (measured by the green space density—Green) and land development potentials (details on the calculation of this indicator were provided in [35]; Available construction land of each county—ACLD). With respect to locational variables, we included a variable representing whether a county is a coastal county and two distance variables measuring the geographical proximity of a county to Xiamen (the state-level special economic zone in Fujian province and the most affluent city in terms of per capita GDP) and to its affiliated prefecture-level city. As economic activities were unevenly distributed across places, those places with locational advantages such as being adjacent to the province’s growth center would attract more development opportunities, i.e., the spatial spillover effects well-recognized in the economic geography literature [19,36,37]. These factors could affect the levels of urban development as well as pollution emissions and would compromise the estimated relationships between urban expansion and pollutant emissions if excluded from the modelling analysis. The statistical summaries of variables were displayed in Table 1.

### 2.5. Statistical Modeling

Given the spatio-temporal nature of our data, the study employed a recently developed Bayesian spatio-temporal statistical model to capture the spatial correlations between spatial units and temporal dynamics across time periods. Denoting the non-overlapping counties constituting the study area as *A_k_* (*k* = 1, 2, …, *K*) and time periods as *t* (*t* = 1, 2, …, 5), the statistical model is specified as
(1)$$y_{kt}|\mu_{kt}\;\sim \;f\left(y_{kt}|\mu_{kt},\;\mu_{kt}\right);\;k=1,2,\ldots ,K;t=1,\;2,\;\ldots ,\;5$$
$$\mu_{kt}=X_{kt}\sigma +U_{kt}\gamma +\psi_{kt}$$

In Equation (1), *y_kt_* is the observed pollutant emission of county *k* at time *t*, which follows a normal distribution with mean $\mu_{kt}$ and variance *σ*^2^, N($\mu_{kt}$, *σ*^2^); *U_kt_* are the urban development intensity variables measured from two different data sources, while *X_kt_* are other predictor variables; ***β*** and ***Υ*** are coefficient vectors to be estimated.

The $\psi_{kt}$ term is a latent component for county *k* at time *t*, and they collectively ($\psi =\psi_{1},\psi_{2},\ldots ,\psi_{5}$ where $\psi_{t}=\psi_{1t},\psi_{2t},\ldots ,\psi_{Kt}$) capture the structured spatiotemporal random effects underlying the data. Spatial correlations are modeled by introducing a *K* × *K* spatial weights matrix *W* specifying the potential spatial connection structure among counties. Following the spatial modeling convention [16], each element of *W* was formed on the basis of geographical contiguity: *w_kj_* = 1 if counties k and j share a common geographical border and 0 otherwise. In order to model the spatially and temporally evolved pollutant emissions surfaces, we formulated a Bayesian spatio-temporal autoregressive model following Rushworth et al. (2014, 2017) [17,38],
(2)$$\mu_{kt}=X_{kt}\beta +U_{kt}\gamma +\psi_{kt}$$
(3)$$\psi_{t}|\psi_{t-1}\sim N\left(\lambda \psi_{t-1},\;\tau^{2}\Omega {\left(W,\;\rho \right)}^{-1}\right)$$
(4)$$\psi_{1}\sim \;N\left(0,\;\tau^{2}\Omega {\left(W,\;\rho \right)}^{-1}\right)$$
(5)$$\tau^{2}\sim \;\mathrm{Inverse}-\mathrm{Gamma}\left(a,b\right);\;\lambda ,\;\rho \;\sim \;\mathrm{Uniform}\left(0,1\right)$$

Equation (3) specifies that the K × 1 vector of random effects for time t evolving over time follow a multivariate first-order autoregressive process with a temporal autoregressive parameter *λ.* The distribution of random effects at time period 1 ($\psi_{1}$) is specified in Equation (4) as a Gaussian Markov Random Field (GMRF) model [39]. The spatial autocorrelation among counties is introduced by the precision matrix Ω (*W*, ρ). Following Rushworth et al. [17], we adopted a special conditional autoregressive (CAR) model developed by Leroux et al. [40] in this study. The conditional distribution of random effect of county k at time period 1, $\psi_{k1}$, given other random effects (***ψ***_−***k*1**_), is formulated as [17]
(6)$$\psi_{k1}|\psi_{-k1},\;W,\rho ,\tau^{2}\;\sim \;N\left(\frac{\rho \sum_{k\sim l}\psi_{l1}}{1-\rho +\rho w_{k+}},\frac{1}{\tau^{2}(1-\rho +\rho w_{k+})}\right)\;(2)$$
where *w_k+_* is the number of geographical neighbors that a county *k* has, and ρ measures the strength of spatial correlation. The resulting precision matrix of the LCAR model is: $\Omega_{\mathrm{LCAR}}=\tau^{2}\left(L_{W}-W\right)\;\mathrm{where}\;L_{W}=\;\mathrm{diag}\;\left(1-\rho +\rho w_{+}\right)$. To complete the specification of the Bayesian spatiotemporal autoregressive model, conventional prior distributions were specified for unknown model parameters: a multivariate normal distribution for regression coefficients (*β* and *Υ*); an inverse-gamma distribution for variance parameters (*σ*^2^ and *τ*^2^); and a uniform distribution for spatial and temporal autoregressive parameters (ρ and *λ*). In our empirical study, non-informative priors were used in the model estimation.

The model was implemented by using the Bayesian Markov Chain Monte Carlo (MCMC) simulation approach, available in an open-source R software package *CARBayeST* [18]. For each of the models implemented below, statistical inferences were based on two MCMC chains, each of which consisted of 70,000 iterations with a burn-in period of 20,000 to ensure the convergence of samplers. We further retained every tenth sample to reduce autocorrelation in each MCMC chain. Model convergence was examined by visual inspection. Deviance information criteria (DIC) [41] were used for model comparisons. A simple working flowchart of the present study is presented in Figure 2.

## 3. Results

### 3.1. Spatio-Temporal Dynamics of Pollutant Emissions and Urban Expansion

The spatial distributions of pollutant emissions and urban expansion and their temporal evolutions are presented in Figure 3. In 1995, the lowest amount of PM_2.5_ emissions was recorded in Siming District (1.857, on the log scale) of Xiamen, while the highest emissions were recorded in Gulou District (6.083) of Fuzhou. Siming District again emitted the smallest amount of PM_2.5_ emissions in 2015, while Taijiang District of Fuzhou became the area emitting the largest amount of PM_2.5_. Overall, there were 19 counties experiencing a decrease in PM_2.5_ emissions from 1995 to 2015, with the largest decrease being Pingnan County. By contrast, there were more than 77% of counties in Fujian province that experienced increases in PM_2.5_ emissions with an average increase of 0.104 in PM_2.5_ emissions during the study period. It was also clearly seen that spatial clusters of relatively high PM_2.5_ emissions gradually formed, surrounding Siming and Haicang Districts of Xiamen and Gulou and Taijiang Districts of Fuzhou. The spatial patterns of NOx emissions and their temporal evolution were largely similar to those observed for PM_2.5_ emissions (maps and statistics are available upon request).

With respect to urban expansion, the lowest ULDI value was recorded for Jianning County of Sanming City in 1995, while the highest level of urbanization was found in the Taijiang District of Fuzhou. The county with the lowest ULDI value in 2015 became Zherong County of Ningde, whilst Taijiang District was still associated with the largest ULDI value. From 1995 to 2015, urban land development levels were increased by about 60% on average for all counties of Fujian province. The spatial patterns of urbanization in 1995 and 2015 measured by the NTL data were very similar to those revealed by the ULDI indicator (Figure 3). The similarity in the spatial distributions of PM_2.5_ emission and urban expansion suggested a potentially positive correlation between them, which will be further tested in our statistical models.

### 3.2. Model Estimation Results

We implemented a series of models to explore the relationships between pollutant emissions (PM_2.5_ and NO_x_) and urban land development. We first note that both the spatial and temporal autoregressive parameters are very high (both > 0.9) with narrow 95% credible intervals, indicating significant spatial correlations and temporal dependences underlying the data and signifying the necessity of a spatio-temporal autoregressive statistical modeling of pollutant emissions. Urban expansion was statistically significantly associated with higher PM_2.5_ and NO_x_ emission intensity (emission per km^2^). Based on the estimated regression coefficients (Table 2), an increase of 0.05 in ULDI (roughly the average increase of ULDI from 1995 to 2015) was associated with about 7.9% (with a 95% credible interval of [5.3%, 9.9%]) increases in PM_2.5_ emissions and about 8.8% (with a 95% credible interval of [6.1%, 10.8%]) increases in NO_x_ emissions, ceteris paribus. Previous national-scale studies found that China’s urbanization process helped to reduce pollutant emissions and concentrations partly because of the higher energy use efficiency in urban areas made available to people migrating from rural areas with lower energy use efficiency and a poor energy mix [26]. Such seemingly contradictory findings would require a further multiple-scale analysis of how pollutant emissions respond to urban expansion and its geographical scale dependency. Nonetheless, unlike the univariate analyses in [26], this study adopted a more suitable spatio-temporal dynamic model and considered potential confounding effects.

With respect to other covariate effects, we found that locational factors were statistically significantly related to PM_2.5_ and NO_x_ emissions: counties further away from the Economic Special Zone (ESZ) Xiamen and the affiliated prefecture city were associated with lower pollutant emission intensity, holding other variables constant. A plausible explanation is that counties far from the economic growth hotspots such as the ESZ Xiamen tend to have fewer development opportunities (e.g., domestic and foreign direct investment). Green land of a county appeared to be negatively correlated with pollutant emission intensity, but this association was not statistically significant at the 95% credible interval. Available construction land of a county and being a coastal county were not statistically significantly associated with pollutant emission.

We then turned to test whether the relationships between urban expansion and pollutant emissions depended on geographical locations of counties. A series of interaction terms between urban expansion and locational factors were sequentially added to the above model (Table 3). The statistically significant interaction term was the one between urban expansion and whether a county is coastal. The negative coefficient of the interaction term highlighted that urban expansion had a much smaller effect on pollutant emissions in coastal areas than in inland areas. High economic development levels and tight environmental regulations implemented in coastal areas might contribute to the above findings, but rigorous tests on causes of the differential effects are left for future research. We note that estimates on other model parameters remained similar to those in Table 2.

In order to obtain reliable estimates on the relationship between urban expansion and pollutant emissions, we implemented models with urban expansion measured by the NTL data. Model estimation results for PM_2.5_ and NO_x_ emissions were reported in Table 4. Estimates of the coefficients of LD were both statistically significant, implying that urban expansion was positively related to pollutant emissions. An increase of 0.04 in LD (roughly the average increase of LD from 1995 to 2015) was associated with about 2.5% (with a 95% credible interval of [1.6%, 3.3%]) increases in PM_2.5_ emissions and about 3.2% (with a 95% credible interval of [2.3%, 4.1%]) increases in NO_x_ emissions, ceteris paribus. We note that the magnitudes of the estimates on how LD and ULDI were related to pollution emissions intensity were different. However, these estimates were consistent in terms of suggesting that urban expansion was associated with increasing pollutant emissions intensity in the study area. Locational factors still presented statistically significant associations with pollutant emissions as found in the above models. We also added meteorological factors such as the average wind speed of each county in the pollution emission models, but found they were not statistically significantly related to pollutant emissions intensity. For instance, the coefficients of the wind speed variable in the PM_2.5_ and NO_x_ model were 0.543 [−0.206, 1.547] and 1.135 [−0.27, 2.245].

Figure 4 presents model-based estimates on the temporal evolutions of PM_2.5_ and NO_x_ emissions from 1995 to 2015, superimposed by the lower and upper quantiles of these estimates (dashed lines). Two interesting features were revealed. First, the overall pollutant emission intensities (PM_2.5_ and NO_x_) were bouncing back and forth with time, rather than presenting a steady trend of increase or decrease. In addition, the gap or variability in pollutant emissions among counties in Fujian Province was shrinking during the two decades. Figure 5 presented the estimates on spatial distributions of PM_2.5_ and NOx, net of covariate effects, in the study area. Spatial correlations of these estimates were clearly shown with hotspots around the Jimei and Xiangan districts of Xiamen and gradually decreasing to the surrounding areas (Figure 5).

## 4. Discussion

How environmental quality responds to urbanization is an important theoretical and empirical enquiry to pursue, given the fast pace of urbanization processes taking place at both global and regional scales. This study proposed a Bayesian dynamic spatio-temporal statistical model to analyze the relationship between urban expansion and pollutant emissions while explicitly capturing the spatial autocorrelation, heterogeneity, and temporal dynamic effects. Quantifications of these effects were interesting in themselves, but more importantly, it allowed for more reliable estimates of the relationship between urban expansion and pollutant emissions [17,18].

Our model estimation results suggest a consistent positive relationship between urban expansion and pollution: urban expansion tended to lead to an increase in pollutant emission intensity in Fujian province during the last two decades. This appears to be in contradiction with the finding that urbanization led to decreases of pollution emissions and concentrations in a recent national-scale study [26]. Meanwhile, it highlights an important issue of scale effects when examining relationships between variables, i.e., the well-recognized modifiable areal unit problem (MAUP) in the spatial analysis and modeling literature [42,43]. Briefly, it refers to the fact that relationships between variables could be very different in magnitudes or even reversed when examined at different spatial scales. This could reflect that the underlying process governing the relationship between two variables could be different at different scales. We therefore argued that at a finer spatial scale (counties in this study), urban expansion could exert detrimental impacts on environmental quality, but at a coarser national scale, urbanization could be beneficial to environmental quality through various channels, such as improved energy production and use efficiency [26]. The potential heterogeneous impacts of urban expansion upon environmental pollution at local and national scales would have important urban and regional development policy implications. One-for-all national uniform policies targeted to urbanization may not be as effective and environmentally friendly as anticipated.

Some limitations remain. First, the results on the relationship between urban expansion and pollutant emissions should not be interpreted as cause and effect. One reason is that certain confounding variables that affect both pollutant emissions and urban expansion might be not incorporated in our model because of data limitations. Second, the relationship between urban expansion and pollution concentrations was not explored in the present study. Although real-time air quality was an optional data source, it lacks the temporal coverage of our emission data. It also suffers from noises added by various spatial interpolation techniques and issues from selective choices of monitoring sites. The next step of our research is to use the GEOS-Chem atmospheric transport model with emission data as key inputs to derive high-resolution pollutant concentration indicators [23,27]. We then will examine the relationships between urban expansion and pollution concentration with the Bayesian spatio-temporal statistical models. Finally, with new credible data sources on urban expansion for other Chinese provinces, we shall test whether the relationship between urban expansion varies across provinces and discuss the potential mechanisms leading to such spatial heterogeneities.

## 5. Conclusions

In this study, we analyzed how pollutant emissions were responding to urban expansion in Fujian province. Our exploration mainly drew upon the fine spatial resolution pollution emissions data and the spatio-temporally matched urban expansion indicators extracted from various remote sensing and satellite data sources. A Bayesian spatio-temporal autoregressive statistical model was developed to examine the relationship between urban expansion and pollutant emissions, explicitly modeling potential spatial correlations and temporal dependency underlying the data. Our empirical results showed that urban expansion in Fujian province elevated pollution emission intensity during 1995–2015. This finding was retained for different measures of urban expansion and after controlling for potential confounding effects. Model-based estimates of the spatial distributions of pollutant emission presented clear clustering patterns. The temporal evolutions of pollutant emission, net of covariate effects, showed a fluctuation pattern rather than a consistent trend of increase or decrease. Nonetheless, the spatial variability of pollutant emissions intensity among counties in Fujian province was decreasing with time.

## Figures and Tables

**Figure 1 ijerph-17-00629-f001:**
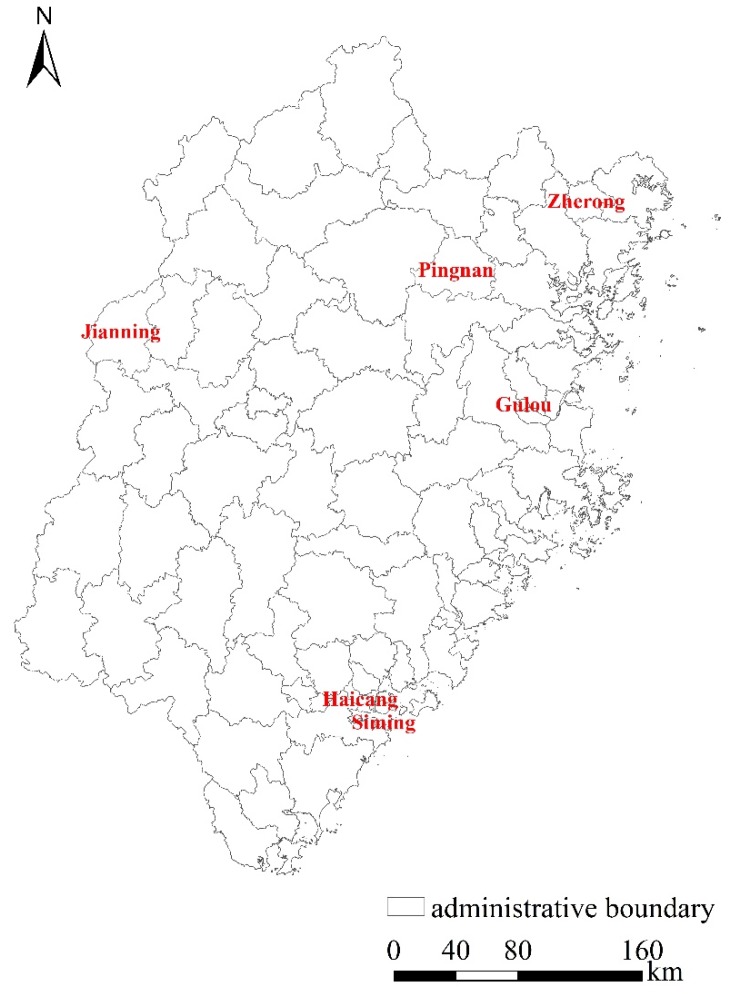
The study area and the county boundaries in Fujian province.

**Figure 2 ijerph-17-00629-f002:**
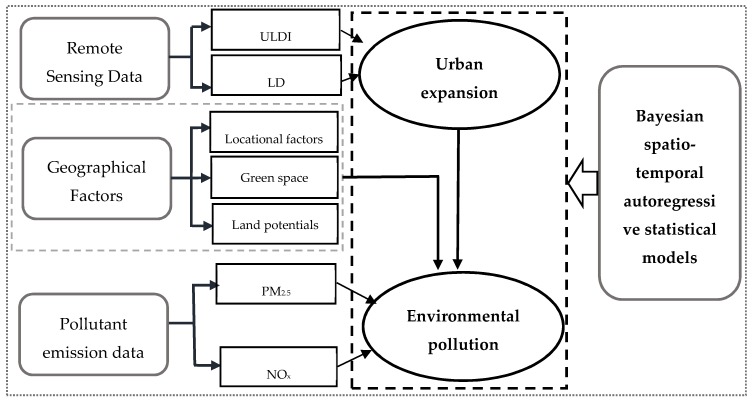
A simple working flowchart of the present study. Note: locational factors in the diagram included a dummy variable representing whether a county is a coastal county and two distance variables measuring the geographical proximity of a county to Xiamen and to its affiliated prefecture-level city.

**Figure 3 ijerph-17-00629-f003:**
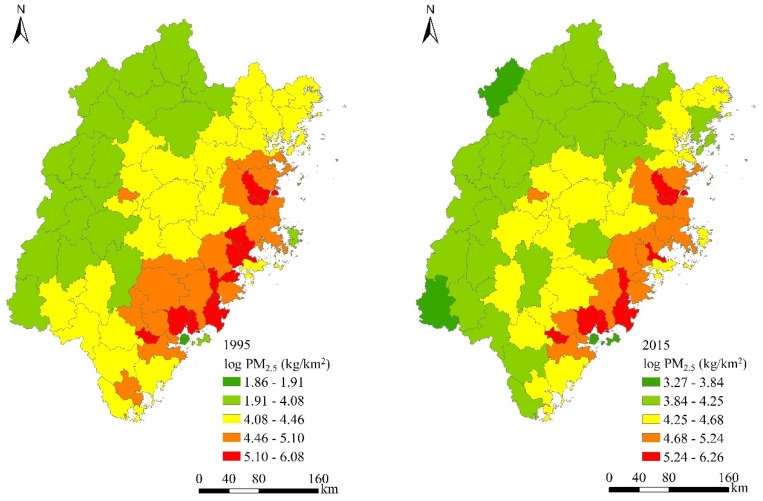
Spatio-temporal dynamics of Log PM_2.5_, ULDI, and LD from 1995 to 2015 in Fujian Province.

**Figure 4 ijerph-17-00629-f004:**
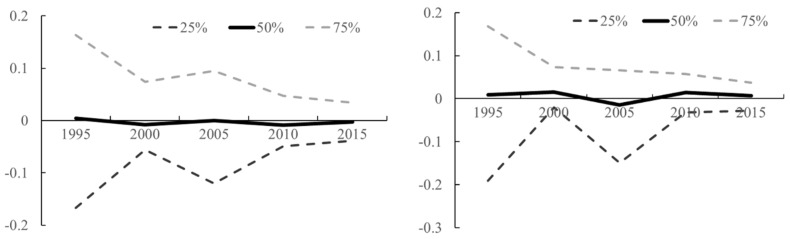
Model-based estimates on the temporal trends of PM_2.5_ (**left**) and NO_x_ (**right**) emissions. The 25%, 50%, and 75% lines represented the lower quantile, median, and upper quantile of model-based estimates on count-level pollution emissions after adjusting for covariate effects during 1995 and 2015.

**Figure 5 ijerph-17-00629-f005:**
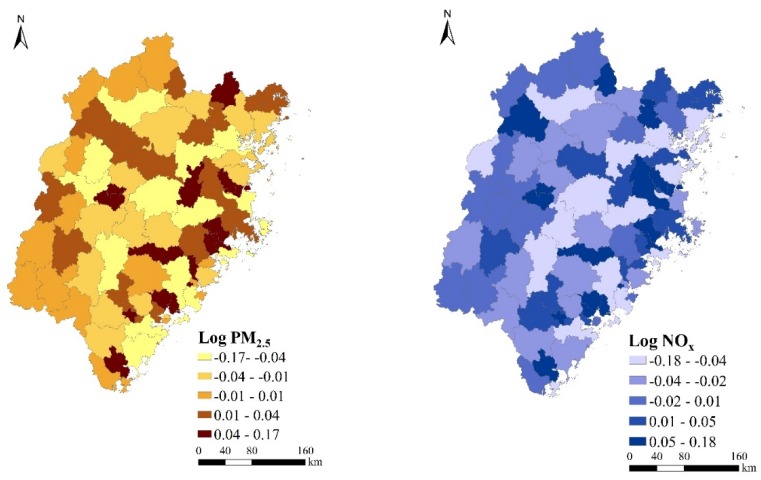
Estimated spatial patterns of PM_2.5_ (**left**) and NO_x_ (**right**) emissions.

**Table 1 ijerph-17-00629-t001:** Statistical summaries on data and variables.

Variables	Description	Mean	Standard Deviation
Log PM_2.5_	Log of PM_2.5_ emission intensity of each county (kg/km^2^)	4.49 (1995)	0.71
		4.59 (2015)	0.63
Log NO_x_	Log of NO_x_ emission intensity of each county (kg/km^2^)	4.66 (1995)	0.65
		5.00 (2015)	0.67
ULDI	urban land development intensity of each county	0.080 (1995)	0.15
		0.128 (2015)	0.21
LD	NTL density of each county	0.102 (1995)	0.16
		0.141 (2015)	0.22
Green	Green space density of each county	0.518 (1995)	0.89
		0.541 (2015)	0.65
ACLD	Available construction land of each county	0.076 (1995)	0.06
		0.067 (2015)	0.05
Dist_City	Log of the distance of each county to its prefecture-level city	1.518	0.60
Coastal	Dummy variable: 1 for coastal county; 0 otherwise	41.2%	---
Dist_Xiamen	Log of the distance of each county to Xiamen (km)	2.136	0.42

ULDI: urban land development intensity; LD: Urbanization was measured by dividing the total luminosity of a county by its total area. ACLD: Available construction land of each county.

**Table 2 ijerph-17-00629-t002:** Estimation results from the dynamic spatio-temporal pollutant emissions model.

Variables	PM_2.5_ Model			NO_x_ Model		
	Median	2.5%	97.5%	Median	2.5%	97.5%
Intercept	11.76 *	8.775	13.73	11.52 *	8.908	14.37
ULDI	1.574 *	1.062	1.984	1.764 *	1.215	2.155
Green	−0.021	−0.026	0.066	−0.026	−0.025	0.077
ACLD	1.448	−2.191	4.705	1.079	−1.72	4.502
Coastal	0.044	−0.152	0.262	−0.056	−0.278	0.14
Dist_City	−0.163 *	−0.205	−0.114	−0.175 *	−0.224	−0.122
Dist_Xiamen	−0.459 *	−0.658	−0.196	−0.485 *	−0.921	−0.215
*τ^2^*	0.293	0.255	0.339	0.349	0.304	0.403
*σ^2^*	0.0021	0.0011	0.0040	0.0027	0.0013	0.0056
ρ	0.981	0.961	0.993	0.935	0.873	0.974
*λ*	0.952	0.898	0.994	0.952	0.899	0.994
DIC	−1044.4			−942.2		
Likelihood-value	902.3			844.7		

Note: the symbol “*” indicates statistical significance at the 95% credible interval. The columns labeled as “Median”, “2.5%”, and “97.5%” reported the median (i.e., the 50th percentile), the 2.5th percentile, and the 97.5th percentile of the estimated distributions of regression coefficients. A regression coefficient is statistically significant at the 95% credible interval if the 2.5th and 97.5th percentiles of the distribution of this parameter do not contain zero. DIC: Deviance information criteria.

**Table 3 ijerph-17-00629-t003:** Model estimation results on the potential spatial heterogeneity effect in the relationship between urban expansion and pollutant emissions.

Variables	PM_2.5_ Model			NO_x_ Model		
	Median	2.5%	97.5%	Median	2.5%	97.5%
ULDI	1.809 *	1.341	2.204	2.120 *	1.537	2.655
Coastal	0.111	−0.070	0.319	0.033	−0.208	0.262
ULDI × Coastal	−1.099 *	−1.981	−0.068	−1.428 *	−2.491	−0.522
Other covariates	Yes			Yes		
*τ^2^*	0.309	0.269	0.356	0.355	0.308	0.410
*σ^2^*	0.0021	0.0011	0.0042	0.0027	0.0014	0.0054
ρ	0.973	0.945	0.989	0.923	0.853	0.968
*λ*	0.957	0.905	0.996	0.957	0.905	0.996
DIC	−1035.3			−932.1		
Likelihood-value	897.7			841.9		

Note: the symbol “*” indicates statistical significance at the 95% credible interval. A regression coefficient is statistically significant at the 95% credible interval if the 2.5th and 97.5th percentiles of the distribution of this parameter do not contain zero. Other covariates included in the previous models (Table 2) were also incorporated in the current model. Estimates on covariate effects were very similar to those reported in Table 2; hence, they were omitted to save space.

**Table 4 ijerph-17-00629-t004:** Model estimation results of models where urban expansion is measured by nighttime light (NTL) data.

Variables	PM_2.5_ Model			NO_x_ Model		
	Median	2.5%	97.5%	Median	2.5%	97.5%
Intercept	10.51 *	6.708	14.18	10.37 *	7.308	14.28
LD	0.629 *	0.411	0.815	0.791 *	0.577	1.034
Green	0.010	−0.041	0.061	0.013	−0.037	0.069
ACLD	1.677	−2.453	4.960	1.424	−2.028	4.836
Coastal	0.031	−0.205	0.227	−0.020	−0.23	0.177
Dist_City	−0.165 *	−0.213	−0.122	−0.178 *	−0.224	−0.127
Dist_Xiamen	−0.417 *	−0.666	−0.068	−0.366 *	−0.748	−0.123
*τ^2^*	0.308	0.268	0.355	0.355	0.309	0.411
*σ^2^*	0.0021	0.0011	0.0041	0.0027	0.0013	0.0055
ρ	0.979	0.956	0.992	0.935	0.874	0.974
*λ*	0.952	0.899	0.994	0.950	0.896	0.993
DIC	−1037.4			−937.9		
Likelihood-value	898.7			843.4		

Note: the symbol “*” indicates statistical significance at the 95% credible interval.

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
