# Peer review of "A Dynamic Spatio-Temporal Analysis of Urban Expansion and Pollutant Emissions in Fujian Province"

_ijerph, 2020, doi:10.3390/ijerph17020629_

Round 1

Reviewer 1 Report

This paper investigated the relationship between urban expansion and pollutant emissions in Fujian province of China, by building a Bayesian spatio-temporal autoregressive model with the data from 1995 to 2015. With reasonable structure and accurate method, this paper is suitable for the magazine. But it could be improved in the following aspects:

In line 99 and 100, why to mention the standard deviation of county area and population. In figure 1, what is the red points (or area) in circle of “Fujian” In line 149, what does the variable of distance of a county to Xiamen mean? Further explain the importance of this variable. In table 1, the values of PM2.5, NOx and SO2 are logarithm already, so what is “proportions” in column 3? And show the max, min, standard deviation values and observations of all variables. What is “one point=0.01” There is not names and signals of counties in figure 2, we cannot understand the analysis combined with the map. Or it could be more intuitive to show the values in tables than dotted figures. please briefly explain the “median 2.5% 97.5” in table2 In line 250, where does “39.8%” come from? The values in text are different with the numbers in table. Please show the calculation when needed for the whole passage. To show the variables of horizontal and vertical coordinates in figure 3. What is the meaning of “25%,50%,75%”?

Author Response

Please see the attached response letter

Reviewer 2 Report

The amount of data is sufficient, and they are effective, traceable and persuasive. The method used is reliable with a theoretical basis and innovations. The research results have certain significance. This article embodies a very complete and meaningful research. I just have some opinions on some small problems in the article:

In the introduction, it is mentioned that the existing research is only with coarse spatial scales, which is also one of the main innovations of this study. In this part, I suggest that the author can mention the scale of these studies, so as to directly reflect the improvement of spatial resolution of the study. If you only look at the research scale of this article, you can't feel that it is a fine scale research.

Although the research is based on county, the resolution of data is also very important. If the resolution of data is not high enough, it will greatly damage the accuracy of results. Therefore, in the part of describing data, I suggest the author to indicate the scale of several most important types of data, or make a table to summarize it. For example, "political emission data were compiled with fine spatial resolution (~ 0.1 by 0.1)" is ok. Most of the data in this paper do not indicate the scale, although many of them refer to the source and the corresponding links.

In Section 3.1, the author describes the info of some place according to graph and statistical results. First of all, I think the key and prominent areas mentioned by the author can be marked on the map (Figure 2), so that the readers can notice the significant changes of these areas directly. Secondly, from Figure 2, we can see some relationship between some pollution emissions and urban expansion. The author can properly discuss and draw preliminary conclusions in 3.1, which also supports the model’s results in the following part.

In the discussion part (After 3.2, including 3.2), the author can add some results of previous studies (such as the article mentioned in the introduction), and compare with the research in this paper to prove the new findings of 20-year time scale and high spatial scale research. Because since there are similar studies (although with different scales), the general direction of conclusions should be mutually supportive, or explore different points, so as to further explain the significance of the study.

there are some typos in this article, I think author should read the whole paper and check again, For example, In the last fourth row of the full text,“The nest? step of our research is to use the GEOS-Chem atmospheric”

Author Response

please see the attached response letter.

Reviewer 3 Report

The paper “A Dynamic Spatio-Temporal Analysis of Urban Expansion and Pollutant Emissions in Fujian Province” analyzes the correlation between urban sprawl and pollutant emissions in the province of Fujian in China. The analyses have been developed adopting the Bayesian spatio-temporal autoregressive model. It could be important to explain what are the main models useful in analyzing this correlation highlighting why the authors decided to use this model and what the main advantages.  

For the sake of clarity in the methodology description it could be better include a flowchart able to synthetize the whole procedure.  

Considering that NOx is also produced by traffic jam it could important to cross the analysed data with modal distribution of transport.

I suggest to separate results and discussion in two different paragraphs. I suggest also to extend the discussion, at this moment it is very poor.  

Author Response

(The authors gave the same response as above.)

Round 2

Reviewer 1 Report

This version is well improved, and could be acceptable.

Author Response

We much appreciate the reviewer's encourage. We have carefully checked the revised manuscript so that no grammatical errors, to the best of our knowledge, exist.

Reviewer 3 Report

Authors accepted great part of my suggestions. The paper can be published.  I encourage the authors to include in the second paragraph a flowchart able to synthetize the whole procedure. 

Author Response

We much appreciate the reviewer's comments. Following the suggestion, we have added a simple working flowchart of the study in the revised manuscript (Figure 2 on Page 7).